# Configurations of Adult Attachment, Indicators of Mental Health and Adverse Childhood Experiences in Women: A Cross-Sectional Study

**DOI:** 10.3390/ijerph182413385

**Published:** 2021-12-19

**Authors:** María Dolores Méndez-Méndez, Yolanda Fontanil, Yolanda Martín-Higarza, Natalia Fernández-Álvarez, Esteban Ezama

**Affiliations:** 1Central University Hospital of Asturias, Mental Health Services of the Principality of Asturias, 33011 Oviedo, Spain; 2Department of Psychology, University of Oviedo, 33003 Oviedo, Spain; fontanil@uniovi.es (Y.F.); fernandezanatalia@uniovi.es (N.F.-Á.); 3Institute of Legal Medicine, Government of the Principality of Asturias, 33001 Oviedo, Spain; mariayolanda.martinhigarza@asturias.org; 4Cabueñes University Hospital, Mental Health Services of the Principality of Asturias, 33201 Gijón, Spain; esteban.ezama.coto@gmail.com

**Keywords:** attachment, mental health, adverse childhood experiences, psychopathology, satisfaction with life, positive affect, negative affect

## Abstract

The relationship between adverse childhood experiences, attachment and adult mental health has been pointed out in a large amount of studies. In a sample of 339 women receiving support from mental health and social services, this research analyzed the association between three adult attachment variables (fear of rejection or abandonment—FRA; desire for closeness—DC; preference for independence—PI) and four mental health indicators. After dichotomizing these variables, we constructed eight configurations of attachment and examined their association with mental health indicators. BAB people (those below the median in FRA, above in DC and below in PI) obtained the most favorable scores in mental health, whereas the ABA configuration (above the median in FRA, below in DC and above in PI) was the least favorable. The association between attachment configurations and mental health indicators was different to what might be expected, aggregating the effects of individual attachment variables. When analyzing the relationship between configurations and adverse childhood experiences (ACEs), women with an ABA configuration reported the highest number of ACEs and eight ACE types had a higher-than-expected contingency coefficient. In conclusion, these findings suggest that certain adult attachment configurations are associated with a greater number of ACEs and poorer mental health indicators in adult women.

## 1. Introduction

John Bowlby [1] defined attachment as the predisposition or universal human need to form affective relationships that can be drawn upon in times of stress. His theory, which conceives of human beings from within a relational framework, considers the drive to maintain close bonds with others as fundamental [2]. Attachment is a complex affective and behavioral system that comprises the set of strategies deployed from infancy in order to achieve the necessary proximity and care to ensure survival. Although these strategies can be modulated by the influence of significant experiences during adolescence or adulthood [1,3], numerous studies argue that many of them tend to be maintained across the lifespan and even transmitted intergenerationally [4]. Attachment bonds are formed in interactions with caregivers through the development of relationship representations, which Bowlby named internal working models. These schemas, which incorporate expectations about the availability of significant others to provide care and about one’s own capacity to receive care, condition an individual’s concept of self, others and the world; they function as heuristics to guide perceptions and responses in interpersonal relationships throughout the lifespan [5,6].

The theory that attachment styles should be seen as dimensional rather than categorical has been gaining ground ever since research into adult attachment began [7]. Two dimensions, anxiety (fear of rejection or abandonment) and avoidance (discomfort with closeness or discomfort depending on others), have emerged as important variables and form the basis of the four classic adult attachment styles: the secure style (low anxiety and low avoidance), preoccupied style (high anxiety and low avoidance), dismissing style (low anxiety and high avoidance) and fearful style (high anxiety and high avoidance) [8,9]. Although the dimensional perspective has tended to assess adult attachment using two variables, anxiety and avoidance, this has not always been the case. Other authors have proposed three-dimensional models (security, anxiety and avoidance; dependence, anxiety and closeness; fear of rejection, desire for closeness and preference for independence) [10,11,12]. This work develops closer to three-dimensional approaches; it explores the importance of possible configurations of variables in people’s mental health. Explanatory models introduce these three variables in isolation [7,13], an approach that has uncovered strong links between fear of rejection or abandonment and manifestations of psychological dysfunction [14,15,16,17]. Crucially, what is missing is an approach that takes into account the coexistence of multiple expectations and preferences in interpersonal relationships. In other words, it is important to look at configurations, rather than variables in isolation.

We will use three variables, fear of rejection or abandonment, desire for closeness and a preference for independence, in line with Fontanil and colleagues [12]. In order to observe how these preferences are combined in the attachment relationships of each person, we sought to find out whether the combinations between the dimensions can provide new information in predicting the presence of psychic dysfunctions.

Early attachments are considered the basis of socioaffective development, which in turn is associated with psychological well-being and life satisfaction [1,18,19]. Adverse environments characterized by neglect or deprivation may interfere with emotion regulation and the acquisition of skills needed for appropriate psychosocial development [20]. Attachment types, as manifestations of internal working models developed as a way to adapt to parenting styles, can give rise to negative expectations about oneself and one’s relationship with others; if they persist into adulthood, these expectations contribute to the emergence of psychological dysfunction, negatively influencing an individual’s capacity to adapt to adversity [21,22,23]. Conversely, when these internal models produce favorable expectations about one’s own capacity to receive care, they can buffer the impact of adversity [24,25,26]. Given the influence of interpersonal relationships on psychological well-being and adaptive capacity throughout the lifespan, it is important to study which patterns of affective attachments are associated with better or worse mental health.

The degree of security in attachment relationships has an influence on the manifestation and intensity of internalizing and externalizing symptoms [27,28]. Although attachment insecurity is considered a risk factor for the development of psychological difficulties, it is becoming increasingly clear that the relationship between insecure styles and problems is not so straightforward. In fact, some strategies and characteristics of these styles are advantageous for certain tasks. For example, people with high levels of anxiety tend to be more alert to dangers and are quick to detect them, whereas people with high levels of avoidance are generally quick at adopting self-protective measures [29]. However, the problem is in the repeated and inflexible use of strategies typical of insecure styles, strategies formed in contexts of loss, threat or abandonment. This situation favors the onset and persistence of psychological dysfunction by aggravating adverse situations and making it more difficult to seize opportunities. It is here that the association between attachment insecurity and personality disorders, depression, anxiety, somatization, paranoia, addictive behaviors and behavioral disturbances becomes apparent [22,30,31,32,33,34,35].

In parallel, there has been a growing body of research into the links between adverse childhood experiences (ACEs) and adult mental health [36,37]. It could be argued that studies into the impact of ACEs and those focusing on insecure attachment are in fact offering two facets of the same dysfunctional dynamic. The Adverse Childhood Experiences Questionnaire deals primarily with adversity affecting family life. It is therefore reasonable to assume that “yes” responses to the questionnaire items would be indicative of a serious imbalance in the attachment-caregiving dynamic. If so, ACEs—perhaps some to a greater extent than others—could prompt children to develop models of themselves and others that impede their capacity to seek out and take advantage of effective interpersonal supports [21,23,26,38]. Conversely, interaction with accessible and supportive attachment figures can provide psychological resources to cope with future challenges, with the result that people with secure attachment styles tend to recover more quickly from periods of stress and have a better sense of well-being [13,25]. In summary, insecure attachment bonds in adulthood would appear to be a partial reflection of ACEs, reflecting a person’s coping and emotion regulation strategies that were formed during childhood and that evolved in accordance with the interpersonal resources available up to the present. In a way, it could be said that a person’s current attachment style mediates between childhood experiences and adult mental health [39].

## 2. Materials and Methods

### 2.1. Aims of the Study

This paper seeks to examine the relationship between three characteristic variables of adult attachment (fear of rejection or abandonment (FRA) desire for closeness (DC) and a preference for independence (PI) in close relationships) and certain indicators of mental health, as well as the number of ACEs (objective 1). Based on previous research, we hypothesized that one dimension of adult attachment (fear of rejection or abandonment (FRA)) would be statistically significantly correlated with mental health indicators (lower positive affect, higher negative affect, lower satisfaction with life and higher global severity index of psychopathology), as well as a higher number of reported ACEs. The other dimensions of adult attachment (desire for closeness (CD) and a preference for independence (PI)) would not be statistically significantly correlated with mental health indicators or the number of reported ACEs.

In addition, the study explores the association between the mental health indicators and the three attachment variables, taken as different configurations of adult attachment, rather than as independent attachment dimensions (objective 2). To this end, each participant was assigned to a group based on the configuration of the values according to the three attachment variables. Then the groups were compared against each other to analyze how they differed in terms of the severity of psychopathology, positive and negative affect and life satisfaction, allowing us to determine which configurations were the least advantageous for mental health (objective 3). Objectives 2 and 3 belong to an exploratory phase of the study that was not based on previous research; their aims were to examine the initial situation regarding the different configurations resulting from combining the three attachment variables considered.

We aimed to explore the association between the mental health indicators and the attachment variables taken as different configurations of adult attachment, rather than as independent attachment dimensions (objective 4). The hypothesis was that the association between attachment variables separately and mental health indicators would be the same as the association between configurations of attachment variables and mental health indicators. Finally, we assessed whether our hypothesis that configurations would not differ in the accumulation of adverse childhood experiences was confirmed (objective 5).

### 2.2. Sample and Procedure

The sample was made up of 339 women recruited from the public healthcare system in the Principality of Asturias and Catalonia (Spain). Non-probabilistic sampling was carried out from 2018 to 2020, inclusively. All participants were adults receiving mental health and social services support. The exclusion criteria required participants not to have any cognitive, physical or cultural problem affecting their communication during the assessment process. Professionals from mental health and social services invited women using their services to participate, informing them of the objectives and conditions of the research by means of a written information document. Each participant who took part in this cross-sectional study signed an informed consent form and was assigned a numeric identifier code. The assessment consisted of a self-report form those participants, returned to the various services once completed. The participants’ mean age was 41.76 years (SD = 12.97) and their educational level was middle or high for most of the subjects (medium 41%; college 42.8%). More than half were employed (55.4%), although the proportion of unemployed participants was also high (24.5%). The remaining subjects were students (8%), people who had a disability (8%) and retired (4.1%). The Catalonian Institute of Health (code: CEIC-1998) and the Principality of Asturias (code: 76/19) Ethics Committees for Research with Medicines approved this study. The research was carried out in accordance with the ethical standards of the Helsinki Declaration.

### 2.3. Measures

Symptom Assessment-45 Questionnaire (SA-45), Cronbach’s α = 0.95 [40,41]—a self-report that describes psychopathological symptoms, in which the subject is asked to indicate the degree to which they have experienced the symptom in the previous week, between 0 (not at all) and 4 (a lot or extremely). In the current study, only the global severity index score was considered (GSI, Cronbach’s α = 0.96 in our sample).

The Scale of Preferences and Expectations in Close Interpersonal Relationships (EPERIC), Cronbach´s α = 0.80 [12]—EPERIC has 22 items and uses a 5-point Likert scale ranging from 1 (is nothing like what happens to me) to 5 (is very much like what happens to me). This instrument has three subscales corresponding to three dimensions of adult attachment: a preference for independence (PI) (α = 0.71 in our sample), a desire for closeness (DC) (α = 0.67 in our sample) and fear of rejection or abandonment by attachment figures (FRA) (α = 0.88 in our sample).

The Spanish adaptation of the Positive and Negative Affect Schedule (PANAS) [42,43] was used to assess affect. PANAS consists of two independent subscales, one for positive affect (PA) (Cronbach’s α = 0.88) and another for negative affect (NA) (Cronbach’s α = 0.85). Using a Likert-type scale from 1 (very slightly or not at all) to 5 (very much), the 20 items (10 for each subscale) measure the frequency with which the respondent experienced a list of moods over the past month. In our sample the reliability was α = 0.91 for PA and α = 0.92 for NA.

The Satisfaction With Life Scale (Cronbach’s α = 0.86) [44,45] was used to evaluate overall satisfaction with life (SWL). The Spanish version of this instrument contains five items with five response options, ranging from 1 (strongly disagree) to 5 (strongly agree); Cronbach’s α = 0.88 in our sample.

The Adverse Childhood Experiences Questionnaire [36] is an instrument that explores if the participant suffered some adverse experience during the first 18 years of life. The experiences assessed using this questionnaire are physical and emotional abuse or neglect; sexual abuse; divorce of death of the parents; witnessing domestic violence; substance abuse in the household; having a family member with mental illness or who had attempted or committed suicide; and having a household member who had been in prison. The response scale is dichotomous (0 = No; 1 = Yes) and the global score is the sum of the score in each item (10 items in total). In the present study, the reliability was α = 0.78.

### 2.4. Data Analysis

The supposition of normality was checked using the Kolgomorov–Smirnov test; apart from SWL, all variables differed statistically from a normal distribution. To fulfill objective 1, we used Spearman’s correlation analysis; after dichotomizing the EPERIC variables (depending on whether the score was below (B) or above (A) the median of subjects), we used the Mann–Whitney U test to compare scores for the mental health indicators (SWL, PA, NA, and GSI). We used the Kruskal–Wallis and the Mann–Whitney U tests for objectives 2 and 3. The Mann–Whitney U test compared each attachment configuration group against all the other groups (28 comparisons) to identify any statistically significant differences between groups. For each of the four mental health indicators, we then counted the frequency with which the group of participants in one configuration scored significantly higher than those in other configurations. This yielded an SWL, PA, NA and GSI score for each configuration. Finally, we assigned each score a rank, depending on whether it was comparatively favorable for mental health (i.e., higher level for SWL and PA scores; lower level for NA and GSI scores). So, for each mental health variable, a configuration is assigned rank 1 when it outperforms all other configurations if the variable is favorable (SWL and PA), if it outperforms all but one it is assigned rank 2, if it outperforms all but two it is assigned rank 3, and so on. In the case that the comparison is made on scores for an unfavorable mental health variable (NA and GSI) the value 1 is assigned to the configuration if it is out-scored by all the other configurations, 2 if it is out-scored by all but one, and so on. When several settings outperform or are outperformed the same number of times by the other settings, they are assigned the same rank. This produced an ordering of the configurations based on the advantage of each mental health indicator, which allowed us to find the average of each configuration in the four indicators. For objective 4, we used the Mann–Whitney test and the comparison of each configuration with all the others across the four mental health indicators. As can be seen, in this procedure, the configurations that occupy the most advantageous position (rank 1) will be the ones with the best mental health indicators. Objective 5 followed the same procedure as objective 3, this time in relation to ACEs. For the number of ACEs, we obtained an ordering comparable to the mental health variables and we calculated the contingency coefficients between each adverse childhood experience and each configuration. The statistical analysis was carried out using SPSS version 20.0. 

To facilitate the reading of results, Appendix A: Acronyms List (in Appendix A) can be consulted.

## 3. Results

### 3.1. Descriptive Analysis

Table 1 shows the means, medians and standard deviations of the attachment variables, mental health indicators and the number of ACEs.

Table 2 presents the frequencies of the different categories of ACEs, as reported by our participants.

### 3.2. Association Analysis

As shown in the tables above, the prevalence of adverse childhood experiences in our sample of women who have sought help for social or health problems was high, as found in previous research [36,37].

#### 3.2.1. Objective 1: Association between Adult Attachment Variables, Mental Health Indicators and Number of ACEs

The correlation analysis revealed statistically significant relationships between the FRA attachment variable and all four mental health indicators. Contrary to what was expected in the hypothesis, there was also a statistically significant correlation with the number of ACEs. For the DC attachment variable, there was a significant correlation with SWL, PA and NA; whereas for PI, the correlation was significant with NA only (Table 3).

When we compared, using the Mann–Whitney U test, the dichotomized attachment variables, depending on whether the score was below or above the median, we found significant differences between the below-the-median FRA group and its above-the-median counterpart in all four mental health indicators and in the number of ACEs. The below- and above-the-median DC groups differed in the mean rank for PA and NA. The below-the-median PI group differed from its high counterpart in NA and the number of ACEs (Table 4).

As can be seen in Table 4, all the comparisons made with the mental health indicators and the number of ACEs were significant for fear of rejection or abandonment. In contrast, the desire for closeness only shows significant differences in the comparisons made with the affective state of the women in the sample (PA and NA). A preference for independence is significant with respect to both negative affect and the number of adverse childhood experiences, and is higher for women who score above the median.

#### 3.2.2. Objective 2: Association between Adult Attachment Configurations and Mental Health Indicators and ACEs

Combining the group membership variables with the scores below (B) and above (A) the median for the FRA, DC and PI attachment variables, respectively, we obtained eight subject groups:

1-BBB (below the median in FRA, below the median in DC, below the median in PI; N = 53, 15.6%).

2-BBA (below the median in FRA, below the median in DC, above the median in PI; N = 46, 13.6%).

3-BAB (below the median in FRA, above the median in DC, below the median in PI; N = 44, 13%).

4-BAA (below the median in FRA, above the median in DC, above the median in PI; N = 32, 9.4%).

5-ABB (above the median in FRA, below the median in DC, below the median in PI; N = 40, 11.8%).

6-ABA (above the median in FRA, below the median in DC, above the median in PI; N = 35, 10.3%).

7-AAB (above the median in FRA, above the median in DC, below the median in PI; N = 56, 16.5%).

8-AAA (above the median in FRA, above the median in DC, above the median in PI; N = 33, 9.7%).

We next used the Kruskal–Wallis test to see if the mean ranks of the scores from the eight groups did not differ from what is expected by chance for the mental health indicators (SWL, PA, NA and GSI) and ACEs. Contrary to the hypothesis, they did (Table 5).

#### 3.2.3. Objective 3: Comparison between Adult Attachment Configurations in Mental Health Indicators

Having found that attachment configurations differ significantly on mental health indicators and adverse childhood experiences, we then compare them two-by-two to see where the differences come from, and which ones are associated with better mental health outcomes. Mann–Whitney U tests revealed which of the attachment variable configurations were linked to better mental health outcomes (Table 6).

Examining Table 6, we note the following results, which show the position of each attachment configuration regarding the mental health indicators:

Satisfaction with life—The 1-BBB configuration showed significantly more favorable scores in comparison with 6-ABA (5th comparison) and 7-AAB (6th); the same was true for 2-BBA in relation to 6-ABA (11th); scores for 3-BAB were significantly more favorable than 1-BBB (2nd), 2-BBA (8th), 5-ABB (15th), 6-ABA (16th), 7-AAB (17th), and 8-AAA (18th). Finally, scores for 4-BAA were more favorable than 6-ABA (20th) and 7-AAB (21st).

Therefore, regarding satisfaction with life:

1st The 3-BAB pattern is the most advantageous, obtaining the highest mean rank in six of the seven comparisons.

2nd In second place are the 1-BBB and 4-BAA groups, which both achieved the highest mean rank in two of the comparisons.

3rd The 2-BBA pattern had the highest mean rank in one comparison.

The 5-ABB, 6-ABA, 7-AAB and 8-AAA groups did not achieve the highest mean rank in any of the comparisons.

Positive affect—the women with 1-BBB and 2-BBA configurations had significantly more favorable scores than those with the 6-ABA configuration (5th and 11th comparisons). This was also the case for 3-BAB, which was more favorable than 1-BBB (2nd), 2-BBA (8th), 5-ABB (15th), 6-ABA (16th), 7-AAB (17th) and 8-AAA (18th). The 4-BAA and 5-ABB groups achieved significantly more favorable scores than 6-ABA (20th and 23rd), which, in turn, was significantly less favorable than 7-AAB (26th) and 8-AAA (27th).

Concerning positive affect, the 3-BAB pattern once again obtained the highest rank in six of the comparisons, followed by 1-BBB, 2-BBA, 4-BAA, 5-ABB, 7-AAB and 8-AAA, which were highest in one comparison. Finally, 6-ABA did not score the highest in any of the comparisons.

Negative affect—as shown in Table 6, 1-BBB was significantly more favorable than 5-ABB (4th comparison), 6-ABA (5th), 7-AAB (6th) and 8-AAA (7th). The same was true for 2-BBA in relation to 5-ABB (10th) and 6-ABA (11th). The 3-BAB pattern was more favorable than 2-BBA (8th), 4-BAA (14th), 5-ABB (15th), 6-ABA (16th), 7-AAB (17th) and 8-AAA (18th). 4-BAA was significantly more favorable than 5-ABB (19th), 6-ABA (20th) and 7-AAB (21st). Finally, 6-ABA had significantly less favorable scores than 7-AAB (26th) and 8-AAA (27th).

Consequently, 1-BBB and 3-BAB were the patterns showing the most advantageous positions; in second and third place were patterns 4-BAA and 7-AAB, respectively. The most disadvantageous pattern was 6-ABA.

Global severity index—1-BBB scored significantly more favorably than 5-ABB (4th), 6-ABA (5th), 7-AAB (6th) and 8-AAA (7th). This was also the case for the 2-BBA pattern when compared with 5-AAB (10th), 6-ABA (11th) and 7-AAB (12th). Scores for 4-BAA were significantly more favorable than those of 5-ABB (19th), 6-ABA (20th) and 7-AAB (21st). The score of 5-ABB was significantly more favorable than that of 6-ABA (23rd). The result was the same for 8-AAA when compared with 6-ABA (27th) and 7-AAB (28th). Finally, 3-BAB scored significantly more favorably than all the other patterns (2nd, 8th, 14th, 15th, 16th, 17th and 18th).

In summary, with regard to psychopathology, once again, 3-BAB was the most advantageous attachment pattern, comprising the participants with the lowest levels of distress. 1- BBB, 2-BBA, 4-BAA and 8-AAA were in second place; 5-ABB and 7-AAB were in third place.

The participants with the most disadvantageous configuration were once again 6-ABA women.

Full set of mental health indicators—Table 7 shows, firstly, the number of times each attachment configuration scored significantly differently from another configuration and, secondly, the position assigned to each configuration based on these comparisons, in terms of its advantage on both mental health indicators and the number of adverse childhood experiences identified. In other words, the number of times each attachment configuration has an advantageous position over the others in having better mental health indicators and less accumulation of adverse childhood experience types. To obtain a picture of the advantages of each configuration in the full set of indicators, we calculated the average position for each and found that 3-BAB was the most advantageous pattern, followed by 1-BBB and 4-BAA. At the other extreme, 6-ABA was the least favorable.

#### 3.2.4. Objective 4: Differences between Analyzed Associations with Attachment Dimensions Individually versus Attachment Configurations and Mental Health Indicators

We aimed to explore the differences between the associations of the attachment dimensions with the mental health indicators, and the associations between the attachment configurations and the mental health indicators. We also reflected on whether our knowledge in the field of psychopathology grows when considering the attachment configurations instead of the attachment dimensions. Results related to objective 1 (Table 3 and Table 4) and objective 3 (Table 6) were compared to answer the following question: Does membership of a configuration of preferences and expectations give different information than membership of each of the three groups separately (i.e., below or above the median in FRA, below or above the median in DC and below or above the median in PI) with regard to mental health indicators?

To put it another way, does the influence of a given FRA level vary depending on the DC and PI levels? If it does not, then groups of configurations with an attachment variable that does not lead to significant differences in a mental health indicator should not differ from each other in the mean rank of that variable.

For example, based on the results shown in Table 4, the FRA level is associated with the SWL level, but neither the DC nor the PI score level significantly influences the SWL mean rank. Based on this logic, therefore, the mean rank of the BBB configuration should not be significantly different from that of BBA, BAB or BAA. The same logic would apply to ABB when compared with ABA, AAB and AAA. In other words, only the level of FRA significantly differentiates the groups in terms of the SWL level, meaning that only comparisons between the AXX and BXX configurations should lead to significant differences. The same should be true for the GSI mean ranks. With NA, on the other hand, all comparisons should show significant differences. Finally, the mean ranks of the group scores in PA should not differ in the comparisons between configurations that do not share the same last letter (i.e., comparisons of BBB with BBA, BAB with BAA, ABB with ABA and AAB with AAA) but should differ in all other comparisons. Similar reasoning can be applied based on the Spearman’s correlations presented in Table 3. In this case, only the predictions for SWL should differ, which are the same as those for PA. However, this has not been the case.

Table 6 shows these predictions in the columns marked “E”; where the expected U and Rho results differ; the prediction based on the correlations is given secondly. We counted any discrepancies between the prediction and the statistically obtained result. For the SWL variable, we found that nine comparisons deviated from the expected results when the predictions were based on comparisons using the Mann–Whitney U test (two significant differences that were not expected and seven differences that were expected but not obtained), and there were 13 deviations when they were based on Spearman’s correlations (zero unexpected differences and 13 that were expected but not obtained). For the PA variable, we found one unexpected difference and 13 differences that were expected but not obtained. We did not find any unexpected differences for NA, but there were 11 differences that were expected but not obtained. Finally, there were four unexpected differences for GSI and two differences that were expected but not obtained.

Contrary to our initial hypothesis, this exercise suggests that when predicting the presence of psychopathology, configurations provide more information than attachment variables considered in isolation.

#### 3.2.5. Objective 5: Association between Adult Attachment Configurations and the Number and Type of ACEs

With the same reasoning as in the previous objective of the study, the question to be answered is: Does membership of a configuration of preferences and expectations give different information than membership of each of the three groups separately (i.e., below or above the median in FRA, below or above in DC and below or above in PI) in respect to the association with having experienced adverse childhood experiences? We analyzed the association with the number of ACEs and with the type of ACEs. To ascertain whether women in the various attachment configurations differed in the number of ACEs reported, we used—as in objective 3—the Kruskal–Wallis test and the comparison of all configurations against each other with the Mann–Whitney U test. Contrary to the prediction of the hypothesis, the results in Table 8 show that the 1-BBB pattern obtained significantly more favorable scores than the 6-ABA, 7-AAB and 8-AAA groups. The 2-BBA configuration scored more favorably than 6-ABA. 3-BAB was less favorable than 2-BBA, 6-ABA, 7-AAB and 8-AAA. Finally, 6-ABA had significantly more favorable scores than 4-BAA and 5-ABB. Thus, 1-BBB, 3-BAB, 4-BAA and 5-ABB were the most favorable patterns, followed by 2-BBA, then in third place, 7-AAB and 8-AAA, and finally, the 6-ABA configuration had the highest number of ACEs (Table 5).

Looking at the association (statistically significant contingency coefficient (CC)) between each configuration and each adverse childhood experience, we found that the 1-BBB pattern was significantly less frequent in the group of women who had experienced emotional abuse (CC = 0.169), sexual abuse (CC = 0.113), emotional neglect (CC = 0.128), and exposure to domestic violence (CC = 0.125) (Table 9). The same is true for the 3-BAB pattern, which was less frequent in the group reporting experiences of emotional abuse (CC = 0.169) and in the group with emotional neglect (CC = 0.117). The number of people in the 4-BAA group reporting household alcohol or drug problems was also lower than what might occur by chance (CC = 0.108). The associations were reversed for 6-ABA and 7-AAB. The number of women in the 6-ABA group reporting experiences of emotional abuse (CC = 0.180), physical abuse (CC = 0.128), emotional neglect (CC = 0.160), physical neglect (CC = 0.179), parental separation or death (CC = 0.139), witnessing domestic violence (CC = 0.109), a household member with mental illness or who attempted suicide (CC = 0.122) or incarceration of a household member (CC = 0.128) was higher than would be expected due to chance. The 7-AAB pattern was associated with one ACE, namely, household mental illness or suicide.

## 4. Discussion

The objective of the present study was to explore the relationship between the dimensions of adult attachment, mental health and having experienced ACEs in the past. In line with other studies [14,16,17,39,46], our results indicate that attachment is a factor associated with different mental health outcomes. For the women in our study sample, the association is clear in the correlations and in the comparison between the groups of participants with scores below and above the median. The attachment variable with the most weight was FRA, whereas the one with the least weight was PI. Taken individually, PI was associated only with NA. However, when participants were grouped according to their combination of dichotomized EPERIC variables, the role of DC and PI was shown to be more important than that suggested by the correlations and mean rank comparisons. We found that the groups of women with different attachment configurations differed significantly in SWL, PA, NA and manifestations of psychological dysfunction (i.e., GSI). The 3-BAB group had the most favorable set of scores for the four mental health indicators taken together; 1-BBB was the second best, followed by 4-BAA and 2-BBA. 8-AAA was ranked fifth, followed by 7-AAB, 5-AAB, and finally 6-ABA, which had the least favorable score across all indicators.

The comparison of the mean ranks of the mental health indicators for each configuration supports the idea that the configuration has more to say than its component variables. If we accept that preferences and expectations translate into different strategies and behaviors, it is logical to assume that the desire for closeness does not lead to the same strategies for someone who strongly fears abandonment by their attachment figure when compared with someone who has less fear. The same applies to the preference for independence. Research has shown that people who are uncomfortable with closeness tend to minimize the expression of their affective needs and deactivate their attachment system, whereas people with a more anxious attachment style, characterized by fear of rejection or abandonment, tend to maximize these needs and seek help from others and from available resources [47]; our results highlight a need for new investigations, this time into strategies associated with the combination of different preferences and expectations. When examining the relationship between life satisfaction, affect, and manifestations of psychological dysfunction, the contribution of the DC/PI combination is not to be ignored: 3-BAB is clearly more advantageous than 4-BAA in SWL and PA. The same is true in the less favorable direction, as 6-ABA is associated with much worse levels of mental health than 5-ABB: comparably, it is at a disadvantage by 32.3 points in SWL, 54.01 in PA, 30 points in NA and 37 in the GSI. The exhaustive comparison of all configurations against each other provides convincing evidence to suggest that the configuration as a whole is more relevant than the sum of its parts.

As regards ACEs, in our sample we found a higher frequency than what has been reported in other studies on the general population [36,48,49]. Fontanil and colleagues [17] established an association between the mental health indicators and ACEs, as well as the influence of attachment on this relationship. The present research found that some attachment configurations are not only associated with manifestations of psychological dysfunction, but also with a greater number of adverse childhood experiences. This is the case for the 7-AAB and 8-AAA configurations, and especially with 6-ABA, a pattern consistent with Bartholomew and Horowitz’s (1991) [10] fearful style, which Williams and colleagues [50] have associated with childhood abuse. Bowlby [51] argued that adverse experiences with primary attachment figures make individuals more vulnerable to later adversity because they employ strategies developed to cope with the early trauma, and these often lead to dysfunctional behaviors. Further research should deeply explore the interactions between the attachment configurations and different manifestations of psychological dysfunction, such as somatization, obsession-compulsion, interpersonal sensitivity, depression, phobic anxiety, paranoid ideation and psychoticism. Future studies should also analyze whether the attachment configurations are associated with certain affects, for example, anger or despair.

The association that has been found between particular attachment configurations and lower exposure to childhood adversity suggests that ACEs influence the types of strategies which individuals use to establish relationships with others throughout life. In our view, people deal with ACEs by using the knowledge, skills and preferences that they are developing or have already developed. They also make use of the scaffolding that comes from the actions of their social network, especially those of their attachment figures. The expectations and the procedural and declarative knowledge developed to adapt to childhood and adolescent adversity influence a person’s later attempts to overcome difficulties and seize opportunities for adaptation in response to adverse situations that pose a threat to quality of life [16]. 5-ABB, a disadvantageous mental health configuration in our sample of women, occupied a favorable position in terms of the number of ACEs reported. This evidence suggests that this configuration may be associated with a greater ability among individuals to forget or defensively suppress the memory of adverse childhood experiences. To verify this hypothesis, future studies should analyze the association between configurations, coping styles and emotion regulation strategies. If correct, one would expect to find higher avoidance and suppression scores among people with this configuration. Certain coping or emotion regulation strategies may not only explain the mental health quality of the women in the sample but may also indicate their degree of ease in reporting ACEs [17].

This study shed light on the role of different attachment configurations in mental health outcomes and their relationship with early adversity. Despite this, the study has some limitations.

First of all, the research design was cross-sectional and retrospective, so the data analyzed refer to past experiences. Although the use of this methodology is common in research on ACEs, it is not possible to contrast casual or predictive relationships between variables. However, this limitation is not particularly severe for some of the research objectives, as these were exploratory aims in which attachment configurations were used for the first time.

On the other hand, the participants were recruited from social and mental health services, so it is not possible to guarantee the representativeness of the sample and results cannot be generalized. Future studies could replicate the study in a larger sample of the general population for a contrast with our results. It would also be interesting to study gender differentials by introducing a sample of men.

Further research should deeply explore the relationship between the attachment configurations and the strategies employed by people to deal with adversity, such as coping style and emotion regulation.

Both in research and professional practice, there is an increasing interest in the issue of adverse childhood experiences, given its valuable role in prevention and intervention in health assistance. Evidence on the association between ACEs and psychic dysfunctions supports the idea that the problems reported by people in the present are related to vital accumulated exposure to adversity. Hostile circumstances influence people´s ability to cope with their vital tasks, especially those related to help-seeking behavior and the establishment of secure interpersonal relationships.

The results of the present study show that the strategies learned in adverse contexts are associated with lower psychological health, supporting the need for a change in psychotherapeutic interventions. Therapy should enhance the exploration of the past experiences of adversity and the understanding of how these experiences are connected to the current problems of the patient and their difficulties in seeking help and regulating the suffering.

Furthermore, the need to promote secure relationships to protect mental health has been confirmed. In therapeutic contexts, this would be achieved by helping people to build relationships in which they feel that they will not be abandoned or rejected, that they may need other people and that closeness to other human beings is a safe place.

To protect mental health, it is essential to detect and reduce any kind of mistreatment, negligence or household dysfunction during childhood and adolescence and take attachment styles and coping strategies as relevant therapeutic objectives when trying to improve well-being.

## 5. Conclusions

The results of this study converge with previous evidence showing that attachment is associated with mental health outcomes. Our study pointed out that a configuration of attachment variables, as a whole, is more relevant than its parts in the explanation of results in mental health indicators. Concerning the ACE scores, our sample reported a higher frequency than other investigations based on the general population, and we found that certain attachment configurations were also associated with higher number of these experiences. We found that 3-BAB was the most advantageous configuration; nevertheless, 6-ABA was associated with worse outcomes in mental health and a high number of ACEs reported.

In summary, our data support the idea that, for users of mental health and social services, different attachment configurations are differentially associated with recognized ACEs and mental health indicators. What remains to be seen is whether awareness of this relationship by women and professionals could help them to make changes resulting in more efficient strategies for dealing with adversity and seizing life’s opportunities.

## Figures and Tables

**Table 1 ijerph-18-13385-t001:** Descriptive statistics of attachment variables, mental health indicators and number of adverse childhood experiences.

	Mean	Median	SD	Min.	Max.
Fear of rejection or abandonment (FRA)	2.77	2.73	0.99	1	4.91
Desire for closeness (DC)	2.90	2.83	0.84	1	5
Preference for independence (PI)	3.64	3.80	0.90	1	5
Satisfaction with life (SWL)	2.81	2.80	1.05	1	5
Positive affect (PA)	2.68	2.60	0.85	1	5
Negative affect (NA)	2.90	2.90	1.07	1	5
Global Severity index (GSI)	1.44	1.44	0.81	0.4	3.91
Adverse childhood experiences (ACE)	3.58	3.00	2.68	0	10

Note: SD: standard deviations.

**Table 2 ijerph-18-13385-t002:** Frequency and percentage of adverse childhood experiences (ACEs) by type.

Adverse Childhood Experience (ACE) Type	Frequency	%
Emotional abuse	170	50.15
Physical abuse	208	61.36
Sexual abuse	108	31.86
Emotional neglect	191	56.34
Physical neglect	45	13.27
Parental divorce or death	135	39.82
Witnessing domestic violence	96	28.32
Household substance abuse	143	42.18
Household mental disorder	152	44.84
Incarcerated household member	44	12.98

**Table 3 ijerph-18-13385-t003:** Correlations between attachment variables, mental health indicators and adverse childhood experiences.

		FRA	DC	PI
Satisfaction with life	Rho	−0.390 **	0.152 **	−0.039
	Sig.	0.000	0.005	0.476
Positive affect	Rho	−0.317 **	0.202 **	−0.016
	Sig.	0.000	0.000	0.771
Negative affect	Rho	0.463 **	−0.128 *	0.111 *
	Sig.	0.000	0.019	0.040
Global Severity Index	Rho	0.569 **	−0.055	0.077
	Sig.	0.000	0.309	0.159
Number of ACEs	Rho	0.255 **	−0.028	0.101
	Sig.	0.000	0.607	0.063

Note: FRA: fear of rejection or abandonment; DC: desire for closeness; PI: preference for independence; ACEs: adverse childhood experiences; Sig.: significance; **: Significance level ≤ 0.01; *: Significance level ≤ 0.05.

**Table 4 ijerph-18-13385-t004:** Comparison of the mean ranks of the mental health indicators for the low- and high-FRA, DC, and PI Groups.

	FRA		DC		PI	
	Group N = 175 vs. N = 164	MR	Group N = 174 vs. N = 165	MR	Group N = 193 vs. N = 146	MR
SWL	Below M	195.99	Below M	160.30	Below M	176.01
	Above M	142.27	Above M	180.22	Above M	162.06
	M-W U	9802.0 **	M-W U	12,668	M-W U	12,930
	Sig.	0.000	Sig.	0.061	Sig.	0.194
PA	Below M	190.07	Below M	153.88	Below M	175.72
	Above M	148.58	Above M	187.00	Above M	162.45
	M-W U	10,837.0 **	M-W U	11,550 **	M-W U	12,986
	Sig.	0.000	Sig.	0.002	Sig.	0.217
NA	Below M	136.06	Below M	180.70	Below M	159.72
	Above M	206.22	Above M	158.72	Above M	183.59
	M-W U	8410.5 **	M-W U	12,493 *	M-W U	12,105.5 *
	Sig.	0.000	Sig.	0.039	Sig.	0.026
GSI	Below M	127.41	Below M	172.32	Below M	161.78
	Above M	215.45	Above M	167.55	Above M	180.87
	M-W U	6896.5 **	M-W U	13,951	M-W U	12,502
	Sig.	0.000	Sig.	0.654	Sig.	0.076
ACEs	Below M	150.97	Below M	171.46	Below M	157.22
	Above M	190.30	Above M	168.46	Above M	186.90
	M-W U	11,020.5 **	M-W U	14,100.5	M-W U	11,621.5 **
	Sig.	0.000	Sig.	0.776	Sig.	0.005

Note: Below M: below the median; Above M: above the median; M-W U: Mann–Whitney U test; FRA: fear of rejection or abandonment; DC: desire for closeness; PI: preference for independence; SWL: satisfaction with life; PA: positive affect; NA: negative affect; GSI: global severity index; ACEs: adverse childhood experiences; MR: mean rank; Sig.: significance; **: Significance level ≤ 0.01; *: Significance level ≤ 0.05.

**Table 5 ijerph-18-13385-t005:** Comparison of the eight subject groups based on configurations of preferences and expectations in close interpersonal relationships.

		Subject Groups								Kruskal-Wallis Test	
		1-BBB	2-BBA	3-BAB	4-BAA	5-ABB	6-ABA	7-AAB	8-AAA	χ^2^	Sig.
SWL	MR	183.85	173.79	233.99	195.75	150.58	118.04	141.19	159.73	38.85	0.000
PA	MR	165.53	177.86	231.25	191.67	156.16	102.11	155.69	176.61	38.14	0.000
NA	MR	137.1	159.4	101.18	148.73	210.56	240.59	190.81	190.64	58.76	0.000
GSI	MR	125.69	151.58	87.7	150.11	200.76	237.7	226.29	191.26	85.48	0.000
ACEs	MR	138.57	175.54	131.78	162.59	162.91	225.69	190.78	185.17	27.865	0.000
Number of subjects		53	46	44	32	40	35	56	33	Total = 339	

Note: SWL: satisfaction with life; PA: positive affect; NA: negative affect; GSI: global severity index; ACEs: adverse childhood experiences; MR: mean rank; Sig.: significance.

**Table 6 ijerph-18-13385-t006:** Significant differences obtained with the Mann–Whitney U test when comparing each configuration with all others.

Comparison			SWL		PA		NA		GSI	
			E	O	E	O	E	O	E	O
1st	1-BBB	2-BBA	ns	ns	ns	ns	sig	ns	ns	ns
2nd	1-BBB	3-BAB	ns/sig	sig (0.008) * BAB = 57.25 BBB = 42.15	sig	sig (0.002) BAB = 58.91 BBB = 40.77	sig	ns	ns	sig (0.019) * BBB = 55.08 BAB = 41.67
3rd	1-BBB	4-BAA	ns/sig	ns *	sig	ns *	sig	ns *	ns	ns
4th	1-BBB	5-ABB	sig	ns *	sig	ns *	sig	sig (0.001) ABB = 58.05 BBB = 38.66	sig	sig (0.000) ABB = 59.88 BBB = 37.28
5th	1-BBB	6-ABA	sig	sig (0.002) BBB = 51.45 ABA = 33.97	sig	sig (0.006) BBB = 50.6 ABA = 35.26	sig	sig (0.000) ABA = 59.61 BBB = 34.52	sig	sig (0.000) ABA = 60.69 BBB = 33.81
6th	1-BBB	7-AAB	sig	sig (0.023) BBB = 62.08 AAB = 48.29	sig	ns *	sig	sig (0.003) AAB = 63.78 BBB = 45.73	sig	sig (0.000) AAB = 70.86 BBB = 38.25
7th	1-BBB	8-AAA	sig	ns *	sig	ns *	sig	sig (0.011) AAA = 52.12 BBB = 38.13	sig	sig (0.001) AAA = 55.29 BBB = 36.16
8th	2-BBA	3-BAB	ns/sig	sig (0.001) * BAB = 54.64 BBA = 36.76	sig	sig (0.008) BAB = 52.93 BBA = 38.39	sig	sig (0.002) BBA = 53.76 BAB = 36.86	ns	sig (0.002) BBA = 53.68 BAB = 36.94
9th	2-BBA	4-BAA	ns/sig	ns *	sig	ns *	sig	ns *	ns	ns
10th	2-BBA	5-ABB	sig	ns *	sig	ns *	sig	sig (0.011) ABB = 50.80 BBA = 37.15	sig	sig (0.016) ABB = 50.43 BBA = 37.48
11th	2-BBA	6-ABA	sig	sig (0.007) BBA = 47.11 ABA = 32.97	sig	sig (0.000) BBA = 49.18 ABA = 30.24	sig	sig (0.000) ABA = 51.93 BBA = 32.68	sig	sig (0.000) ABA = 52.37 BBA = 32.35
12th	2-BBA	7-AAB	sig	ns *	sig	ns *	sig	ns *	sig	sig (0.000) AAB = 61.31 BBA = 39.55
13th	2-BBA	8-AAA	sig	ns *	sig	ns *	sig	ns *	sig	ns *
14th	3-BAB	4-BAA	ns	ns	ns	ns	sig	sig (0.026) BAA = 45.11 BAB = 33.69	sig	sig (0.002) BAA = 47.81 BAB = 31.73
15th	3-BAB	5-ABB	sig	sig (0.000) BAB = 52.83 ABB = 31.14	sig	sig (0.001) BAB = 51.32 ABB = 32.8	sig	sig (0.000) ABB = 56.13 BAB = 30.11	sig	sig (0.000) ABB = 58.39 BAB = 28.06
16th	3-BAB	6-ABA	sig	sig (0.000) BAB = 51.42 ABA = 25.64	sig	sig (0.000) BAB = 52.68 ABA = 24.06	sig	sig (0.000) ABA = 56.26 BAB = 27.07	sig	sig (0.000) ABA = 56.56 BAB = 26.83
17th	3-BAB	7-AAB	sig	sig (0.000) BAB = 64.34 AAB = 39.63	sig	sig (0.000) BAB = 63.28 AAB = 40.46	sig	sig (0.000) AAB = 62.66 BAB = 35.02	sig	sig (0.000) AAB = 66.71 BAB = 29.88
18th	3-BAB	8-AAA	sig	sig (0.000) BAB = 46.74 AAA = 28.68	sig	sig (0.013) BAB = 44.45 AAA = 31.73	sig	sig (0.000) AAA = 51.03 BAB = 29.98	sig	sig (0.000) AAA = 54.2 BAB = 27.6
19th	4-BAA	5-ABB	sig	ns *	sig	ns *	sig	sig (0.007) ABB = 42.5 BAA = 29	sig	sig (0.019) ABB = 41.69 BAA = 30.02
20th	4-BAA	6-ABA	sig	sig (0.002) BAA = 41.69 ABA = 26.97	sig	sig (0.000) BAA = 43.36 ABA = 25.44	sig	sig (0.000) ABA = 42.86 BAA = 24.31	sig	sig (0.000) ABA = 42.66 BAA = 24.53
21st	4-BAA	7-AAB	sig	sig (0.016) BAA = 53.19 AAB = 9.54	sig	ns *	sig	sig (0.043) AAB = 48.66 BAA = 37.22	sig	sig (0.000) AAB = 51.73 BAA = 31.84
22nd	4-BAA	8-AAA	sig	ns *	sig	ns *	sig	ns *	sig	ns *
23rd	5-ABB	6-ABA	ns	ns	ns	sig (0.013) * ABB = 43.83 ABA = 31.34	sig	ns *	ns	sig (0.012) * ABA = 44.73 ABB = 32.11
24th	5-ABB	7-AAB	ns/sig	ns *	sig	ns *	sig	ns *	ns	ns
25th	5-ABB	8-AAA	ns/sig	ns *	sig	ns *	sig	ns *	ns	ns
26th	6-ABA	7-AAB	ns/sig	ns	sig	sig (0.008) AAB = 51.80 ABA = 36.71	sig	sig (0.004) ABA = 56.04 AAB = 39.72	ns	ns
27th	6-ABA	8-AAA	ns/sig	ns *	sig	sig (0.001) AAA = 42.39 ABA = 27.06	sig	sig (0.016) ABA = 40.11 AAA = 28.55	ns	sig (0.007) * ABA = 40.73 AAA = 27.89
28th	7-AAB	8-AAA	ns	ns	ns	ns	sig	ns *	ns	sig (0.041) * AAB = 49.3 AAA = 37.7

Note: SWL: satisfaction with life; PA: positive affect; NA: negative affect; GSI: global severity index; E: difference expected based on addition of variable effects; O: difference obtained; ns: no statistically significant difference; sig: significant difference; Additional Numbers: mean rank of each configuration; *: Discrepancy between expected and observed.

**Table 7 ijerph-18-13385-t007:** Advantageous positions of attachment configurations on mental health indicators and number of adverse childhood experiences.

		Subject Groups							
		1-BBB	2-BBA	3-BAB	4-BAA	5-ABB	6-ABA	7-AAB	8-AAA
SWL	NU	2	1	6	2	0	0	0	0
	Advan	2	3	1	2	4	4	4	4
PA	NU	1	1	6	1	0	0	2	1
	Advan	3	3	1	3	4	4	2	3
NA	NU	0	1	0	1	4	6	3	2
	Advan	1	2	1	2	5	6	4	3
GSI	NU	1	1	0	1	4	7	4	2
	Advan	2	2	1	2	*3*	4	3	2
Average position for mental health indicators		2	2.5	1	2.25	4	4.5	3.25	3
ACEs	NU	0	1	0	0	0	5	2	2
	Advan	1	2	1	1	1	4	3	3
Number of subjects		53	46	44	32	40	35	56	33

Note: SWL: satisfaction with life; PA: positive affect; NA: negative affect; GSI: global severity index; ACEs: adverse childhood experiences; NU: number of U tests achieving a significantly higher mean rank; Advan: advantageous position in the comparison.

**Table 8 ijerph-18-13385-t008:** Significant differences obtained with the Mann–Whitney U when comparing the number of adverse childhood experiences in the configurations.

	2-BBA	3-BAB	4-BAA	5-ABB	6-ABA	7-AAB	8-AAA
1-BBB	ns	ns	ns	ns	6-ABA = 57.37 1-BBB = 36 sig = 0.000	7-AAB = 63.27 1-BBB = 46.26 sig = 0.005	8-AAA = 50.86 1-BBB = 38.92 sig = 0.029
2-BBA		2-BBA = 51.53 3-BAB = 39.19 sig = 0.024	ns	ns	6-ABA = 47.69 3-BBA = 35.91 sig = 0.025	ns	ns
3-BAB			ns	ns	6-ABA = 51.17 3-BAB = 31.11 sig = 0.000	7-AAB = 57.85 3-BAB = 41.15 sig = 0.004	8-AAA = 45.68 3-BAB = 33.99 sig = 0.022
4-BAA				ns	6-ABA = 40.11 4-BAA = 27.31 sig = 0.007	ns	ns
5-ABB					6-ABA = 45.69 5-ABB = 31.28 sig = 0.004	ns	ns
6-ABA						ns	ns
7-AAB							ns

Note: ns: no statistically significant difference; sig: significant difference.

**Table 9 ijerph-18-13385-t009:** Statistically significant contingency coefficients (CC) and observed (O) and expected (E) frequencies of the yes/yes boxes from the contingency tables.

	1-BBB	2-BBA	3-BAB	4-BAA	5-ABB	6-ABA	7-AAB	8-AAA
Emotional abuse (1)	CC = 0.169 sig = 0.002 O = 16 E = 26.6	ns	CC = 0.123 sig = 0.022 O = 15 E = 22.1	ns	ns	CC = 0.180 sig = 0.001 O = 27 E = 17.6	ns	ns
Physical abuse (2)	ns	ns	ns	ns	ns	CC = 0.128 sig = 0.018 O = 20 E = 13.5	ns	ns
Sexual abuse (3)	CC = 0.153 sig = 0.004 O = 8 E = 16.9	ns	ns	ns	ns	ns	ns	ns
Emotional neglect (4)	CC = 0.128 sig = 0.018 O = 22 E = 29.9	ns	CC = 0.117 sig = 0.001 O = 15 E = 24.8	ns	ns	CC = 0.160 sig = 0.003 O = 28 E = 19.7	ns	ns
Physical neglect (5)	ns	ns	ns	ns	ns	CC = 0.179 sig = 0.001 O = 11 E = 4.6	ns	ns
Parental divorce or death (6)	ns	ns	ns	ns	ns	CC = 0.139 sig = 0.010 O = 21 E = 13.9	ns	ns
Witnessing domestic violence (7)	CC = 0.125 sig = 0.020 O = 8 E = 15	ns	ns	ns	ns	CC = 0.109 sig = 0.044 O = 15 E = 9.9	ns	ns
Household substance abuse (8)	ns	ns	ns	CC = 0.108 sig = 0.046 O = 9 E = 14.3	ns	ns	ns	ns
Household mental disorder (9)	ns	ns	ns	ns	ns	CC = 0.122 sig = 0.024 O = 22 E = 15.7	CC = 0.109 sig = 0.043 O = 32 E = 25.1	ns
Incarcerated household member (10)	ns	ns	ns	ns	ns	CC = 0.128 Exact sig = 0.030 O = 9 E = 4.5 *	ns	ns

Note: Exact sig: * Fisher exact test, calculated as E < 5; ns: no statistically significant difference; sig: significant difference.

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
