# Peer review of "Configurations of Adult Attachment, Indicators of Mental Health and Adverse Childhood Experiences in Women: A Cross-Sectional Study"

_ijerph, 2021, doi:10.3390/ijerph182413385_

Round 1

Reviewer 1 Report

please see the attached file.

Reviewer 2 Report

Thanks for inviting me to review this interesting study. It is a great and well-organized manuscript. My comments are as follows:

    Some detailed revisions are needed.

  • No value of SD was reported about the participants’ age in the sample and procedure section.
  • Spacing between paragraphs is not consistent. In addition, the spacing in the tables is too large which makes the tables too long, and it is difficult to follow especially when the table takes more than one page, e.g., table 6.
  • Validity test results for each scale are strongly suggested to report in the measurement section.
  • Notes are suggested to add to the current tables, e.g., to explain the meaning of *, **. ***, etc.
  • The sampling method should be explained in detail. Currently, it seems that the authors used the nonprobability sampling method which may bring bias to the generalization to the whole population.

In general, this manuscript is well-written, the results can be greatly supported by the data analyzed. In my points, I do not think it is very innovative to analyze the relationship between adult attachment and mental health. Similar studies are very common. So I strongly suggested the authors add more explanation in the introduction part about what is new in this study. In addition, more discussion should be made to demonstrate the implications to practice.

Round 2
